# Immune Profile of Blood, Tissue and Peritoneal Fluid: A Comparative Study in High Grade Serous Epithelial Ovarian Cancer Patients at Interval Debulking Surgery

**DOI:** 10.3390/vaccines10122121

**Published:** 2022-12-12

**Authors:** Pavan Kumar, Samruddhi Ranmale, Hemant Tongaonkar, Jayanti Mania-Pramanik

**Affiliations:** 1ICMR-National Institute for Research in Reproductive and Child Health, Mumbai 400012, Maharashtra, India; 2P. D. Hinduja National Hospital & Medical Research Centre, Mumbai 400016, Maharashtra, India

**Keywords:** ovarian cancer, tumor microenvironment, NK cell receptors, immune modulation, cognate ligands

## Abstract

High-grade serous epithelial ovarian carcinoma (HGSOC) is an immunogenic tumor with a unique tumor microenvironment (TME) that extends to the peritoneal cavity. The immunosuppressive nature of TME imposes the major challenge to develop effective treatment options for HGSOC. Interaction of immune cells in TME is an important factor. Hence, a better understanding of immune profile of TME may be required for exploring alternative treatment options. Immune profiling of peritoneal fluid (PF), tumor specimens, and blood were carried out using flowcytometry, ELISA, and Procartaplex immunoassay. The frequency of CD56^Bright^NK cells and expression of functional receptors were reduced in PF. Increased activating NKp46+CD56^Dim^NK cells may indicate differential antitumor response in PF. Functional receptors on NK, NKT-like and T cells were reduced more drastically in tumor specimens. Soluble ligands MIC-B and PVR were reduced, whereas B7-H6 was increased in PF. Dissemination of tumor cells contributes to soluble ligands in PF. A differential cytokine profile was found in serum and PF as IL-2, IL-8, IL-15, IL-27, IFN-γ, and GM-CSF were elevated specifically in PF. In conclusion, the differential immune profile and correlation of soluble parameters and NK cell receptors with chemo response score may add knowledge to understand anti-tumor immune response to develop effective treatment modality.

## 1. Introduction

High-grade serous epithelial ovarian carcinoma (HGSOC) is the predominant subtype, which constitutes approximately 70% of epithelial ovarian cancers [1]. It is the most aggressive and fatal among all the malignancies of the female reproductive tract, with poor outcomes. Relapse followed by chemoresistance is very common in HGSOC patients, which lead to unfavorable disease outcome. Disease heterogeneity across subtypes and within a single tumor is one of the major reasons for treatment failure [2]. Ovarian cancer is an immunogenic tumor since 55% of patients have a spontaneous antitumor response. It possesses a unique tumor microenvironment (TME) that extends to the peritoneal cavity and blockage of the lymphatic duct leads to the buildup of ascites or peritoneal fluid (PF). The most researched immune compartment in cancer patients is peripheral immune parameters. Peritoneal fluid and the tumor itself, which forms the TME, are the two additional immune compartments. Multiple reports indicate that the immunosuppressive behavior of TME could be a major challenge to developing an effective treatment modality [3]. The complex interactions between immune cells, non-immune cells, soluble mediators and ovarian cancer cells in TME are important factors during disease initiation and progression [4]. The presence of CD3+ tumor-infiltrating T cells (TILs) was associated with 5-year overall survival of ovarian cancer patients. Overall survival was 38% with the presence of TILs, whereas, for those without TILs, overall survival was only 4.5% [5]. Similarly, tumor-associated lymphocytes (TAL) present in malignant ascites could be another important aspect. However, survival of ovarian cancer patients was not shown to be associated with DCs and T cells in ascites [6]. Additionally, it has been discovered that NK cells with altered phenotypes accumulate in PF and infiltrate tumor tissue [7,8]. During the initial diagnosis, a suppressive immunophenotype was reported in tumor-associated NK cells, with a higher percentage of CD56^Bright^NK cells than peripheral blood, along with altered functional markers such as PD-1 and CD39 in ascites [9]. In contrast, another study reported an equal proportion of both subsets of CD56^Bright^NK and CD56^Dim^NK cells and a reduced level of CD16+NK cells in the peritoneal fluid of HGSOC patients [10]. Furthermore, chemotherapy also modulates the expression of functional receptors, viability, proliferation, and cytolytic functions of NK cells [11,12]. A comparison of the phenotype of immune cells from different sites within the same patient may add valuable information on tumor-immune cell interactions. PF or ascites may reflect these interactions as it contains both tumor cells and immunological components such as immune cells, cytokines, and chemokines [13]. Ovarian cancer rarely disseminates in blood [14], making phenotypic comparisons between peripheral blood lymphocytes, tumor-associated lymphocytes (TALs) from PF, and TILs from tumor tissue valuable.

In the current investigation, matched samples of peripheral blood, peritoneal fluid, and tissue specimens from the same HGSOC patients collected at the time of interval debulking surgery are examined for their immunophenotypes of NK, NKT-like, and T cells, cytokines, and soluble ligands. Comparative studies of circulating lymphocytes, tumor-associated lymphocytes (TALs), and tumor-infiltrating lymphocytes (TILs) may offer an immunological basis of therapy for this deadly disease.

## 2. Materials and Methods

### 2.1. Participant Enrollment

High-grade serous epithelial ovarian carcinoma (HGSOC) patients were enrolled in the study at the time of interval debulking surgery (IDS) for the collection of paired blood, tissue and PF specimens from P.D. Hinduja Hospital and Medical Research Centre, Mumbai, India between 2017 and 2021. The study was conducted following the Declaration of Helsinki and approved by the Institutional ethics committee of ICMR-National Institute for Research in Reproductive and Child Health and P.D. Hinduja Hospital, Mumbai. The informed consent form was signed by each enrolled patient. Blood and PF samples were collected in EDTA vacutainer, whereas tissue specimens were collected in RPMI medium. Serum samples from healthy controls (HC) were used for comparative analysis of soluble ligands and cytokine panel.

### 2.2. Preparation of Tumor-Associated Lymphocytes from HGSOC Patients

Peritoneal fluid (PF) specimens (20 mL) were obtained at the time of interval debulking surgery. The surgeon used the standard procedure for PF tapping and took all the precautions to avoid contamination of PF with the peripheral lymphocytes. PF samples received were centrifuged at 2000 rpm for 10 min to pellet the cells. The supernatant was stored at −80 °C for analysis of soluble ligands and cytokines. Centrifuged cells were washed twice with PBS. Pellet was re-suspended in 500 μL of PBS and used for immunophenotyping.

### 2.3. Single Cells Preparation of Tissue Specimens from HGSOC Patients

Tissue specimens were processed by enzymatic and mechanical digestion to obtain single-cell suspension before staining. Briefly, tissue specimens were cut into small pieces followed by 4 mL of 1 mg/mL Collagenase type IV (Gibco^TM^ Thermo Fisher Scientific, Waltham, MA, USA) treatment for 15 min in a water bath at 37 °C. The cell suspension was washed four times with DMEM at 3000 rpm for 10 min by centrifugation and the supernatant was discarded. Trypsin EDTA (4 mL, 0.05%) (Gibco^TM^ Thermo Fisher Scientific, Waltham, MA, USA) +DMEM (Gibco^TM^ Thermo Fisher Scientific, Waltham, MA, USA) was added to the pellet and incubated at 37 °C for 10 min. DMEM (8 mL) +10% FBS (Gibco^TM^ Thermo Fisher Scientific, Waltham, MA, USA) was added to the sample tube and centrifuged at 3000 rpm for 10 min to pellet the cells. Pellet was re-suspended in DMEM and strained through the 40 μm (Corning, Merck, Darmstadt, Germany) cell strainer. The single-cell suspension was used for the immunophenotyping of receptors.

### 2.4. Surface Staining of Peripheral Lymphocytes, Tumor-Associated Lymphocytes, and Tumor-Infiltrating Lymphocytes for Flow Cytometry

Re-suspended cells from peritoneal fluid and whole blood (150 μL) were immune-stained with fluorescently labeled monoclonal antibodies for 30 min at 4 °C in dark. Stained blood and PF were incubated in FACS lysis buffer (BD Biosciences, San Jose, CA, USA) for 15 min with intermittent vortexing to lyse red blood cells (RBCs) and washed twice with staining buffer (0.02% FBS in PBS). A similar staining method was used for tissue infiltrating immune cells except for the RBC lysis. Samples were acquired immediately on the BD FACS Aria™ Fusion (BD Biosciences, San Jose, CA, USA) flow cytometer. The data were analyzed using FlowJo software version 10.1. The threshold for positive staining was determined using unstained or fluorescence minus one (FMO) control. LIVE/DEAD^TM^ Fixable Violet Dead Cell Stain kit (Invitrogen^TM^, Thermo Fisher Scientific, Waltham, MA, USA) was used to exclude dead cells. The lymphocyte population was gated based on the CD45 expression for blood, PF, and tissue specimens. To identify circulating, tumor-associated, and tumor-infiltrating NK, NKT-like, and T cells, a sequential gating strategy was created based on the CD3 and CD56 expression. NK cells were further divided into CD56^Bright^NK and CD56^Dim^NK cells based on the density of CD56 expression. The expression of functional markers on CD56^Bright^NK, CD56^Dim^NK, NKT-like, and T cells were then evaluated by the percentage of positive cells (Appendix A).

### 2.5. Soluble Cytokine and Ligand Levels in Peritoneal Fluid and Serum

Serum and PF level of cytokines such as IL-2, IL-5, IL-6, IL-8, IL-10, IL-15, IL-27, IFN-У, TNF-α, GM-CSF, and two soluble ligands B7-H6 and Poliovirus receptor (PVR) were determined by using ProcartaplexMultiplex Immunoassay (Invitrogen^TM^, Thermo Fisher Scientific, Waltham, MA, USA). The manufacturer’s instructions for the protocol have been adhered to during the experiment. Briefly, the antibody-coated beads against the different cytokines or ligands of interest were processed in a 96-well plate. A magnet ELISA plate holder was used for the whole process. The beads were washed with wash buffer for 30 s, followed by incubation with serum, PF supernatant, and standards for 1 h. This was followed by washings and the addition of enzyme-linked secondary antibodies against all these cytokines and ligands of interest. After incubation, the plate containing the beads was washed. This was followed by the addition of streptavidin–R-phycoerythrin (SAPE) to capture the enzyme-linked antigen–antibody complex. After 30 min of incubation, the plate was washed. The captured antigen–antibody complex beads present in the plate were analyzed on a Luminex™ instrument (Thermo Fisher Scientific, Waltham, MA, USA) to measure their concentration. The standard curve was drawn using standards of known concentrations from the kit. The concentration of soluble cytokines and ligands was interpolated from the standard curve using the Optical Density (OD) of each sample.

### 2.6. Analysis of the Soluble Level of MICA, MICB, in Peritoneal Fluid and Serum

The manufacturer’s instructions were followed for every ELISA (Thermo Fisher Scientific, Waltham, MA, USA) test. Absorbance was taken by using a BiotekEpoch spectrophotometer (Agilent, Santa Clara, CA, USA) at 450 nm as the primary wavelength. Standards of known concentrations, provided in the kit, were used to plot the standard curve, against which the OD of each sample was used to determine the concentration of these analytes.

### 2.7. Statistical Analysis

GraphPad Prism 9.0 (GraphPad Software, San Diego, CA, USA) was used for all statistical analyses. The Kaplan–Meier survival analysis with the log-rank test was carried out to determine the survival difference between groups. Wilcoxon matched-pairs signed rank test analysis was conducted between paired clinical specimens. A comparison between the unpaired groups was drawn using the Mann–Whitney U-test. Spearman’s correlation matrix was used to determine the correlation between variables. Data are reported as mean ± SEM. *p* < 0.05 were considered to be statistically significant.

## 3. Results

### 3.1. Detailed Patient Characteristics and Association of Fluid Cytology with Progression-Free Survival

Thirty-three women diagnosed with HGSOC were enrolled in the study at the time of IDS to collect paired clinical samples such as peripheral blood, PF, and tissue specimens. The median age of the enrolled patients was 56.5 (Range 32–76) years. The median age of the enrolled thirty-four healthy volunteers was 51 (Range 35–67) years. The performance status of these enrolled HGSOC patients evaluated with ECOG scale was in the range of 0–2 (Table 1). Around 12.1% of patients were diagnosed at an early stage (FIGO II) and 87.8% were diagnosed at an advanced stage (FIGO III and IV). Along with HGSOC, 54.5% of patients had other co-morbidities such as hypertension, diabetes, etc. Based on chemotherapy response, 14 (42.4%) patients had no or minimal response/progressive disease (CRS 1), 16 (48.4%) patients had partial/moderate response to chemotherapy/stable disease (CRS 2), and 3 (0.09%) patients had a good response or near complete resolution of tumor lesions (CRS 3). Pre-chemotherapy serum CA-125 levels of these patients were 665.5 (range 125.4–27,428 unit/mL), which was reduced post-chemotherapy to 37.94 (range 6.9–679.5 unit/mL). Dissemination of tumor cells in the PF was found to determine the prognosis of the disease. Kaplan–Meier survival analysis indicates that the presence of malignant tumor cells in the PF (ASC+) was associated with poor survival of ovarian cancer patients. Median progression-free survival of ASC+ patients was 15 months versus 33 months for ASC- patients, HR 0.56 (95% CI 0.3–1.05, *p* = 0.045) (Figure 1a).

### 3.2. Higher Frequency of NKT-Like and T Cells in Peritoneal Fluid Than Tumor Specimens of HGSOC

The frequency of CD56^Bright^NK cells was low in the PF of HGSOC patients (Figure 1b). We did not find CD56^Bright^NK cells in tissue specimens which were confirmed based on the Median fluorescence intensity (MFI) of CD56 expression. CD56^Dim^NK and NKT-like cells were comparable in the PF and peripheral blood, while tumor specimens had a reduced frequency than their peripheral blood (Figure 1c–d). However, the frequency of T cells was reduced in PF than their peripheral blood. The frequency of T cells was also reduced further in tumor specimens (Figure 1e). Furthermore, we hypothesize that the dissemination of tumor cells may influence the presence of lymphocytes. Therefore, we divided patients into two groups based on fluid cytology status positive (ASC+) or negative (ASC-). We did not observe any significant difference in CD56^Bright^NK cells between ASC+ and ASC-groups. However, we have found that PF positive for disseminated tumor cells (ASC+) had a higher frequency of T cells and a trend of increased CD56^Dim^NK cells (Figure 1f–i).

### 3.3. Reduced Activating Receptors Expression on CD56^Bright^NK Cells in Peritoneal Fluid

Analysis of surface receptors in paired clinical specimens in HGSOC patients revealed that NKG2C, NKG2D, and DNAM-1 positive CD56^Bright^NK cells including KIR3DL1+CD56^Bright^NK cells were reduced in PF than their expression in peripheral blood (Figure 2a–d). The expression of other receptors on CD56^Bright^NK cells was comparable between PF and peripheral blood (Data not shown).

### 3.4. Improved Expression of Activating NKp46 on CD56^Dim^NK Cells in Peritoneal Fluid

NKp46+CD56^Dim^ NK cells were high in PF (Figure 3a). However, DNAM-1+CD56^Dim^ NK cells, on the other hand, were steadily reduced in PF and tissue specimens than peripheral blood (Figure 3b). The frequency of NKG2C, NKG2D, CD161, and KIR2DL2/L3/S3 positive CD56^Dim^NK cells was comparable between peripheral blood and PF, whereas these CD56^Dim^NK cell subsets were low in tissue specimens (Figure 3c–f). KIR3DL1+CD56^Dim^NK cells were low in PF than peripheral blood (Figure 3g). KIR2DL1/S1+CD56^Dim^ NK cells were reduced in tissue specimens than in peripheral blood (Figure 3h). There was no significant difference in the expression of these receptors between the ASC+ and ASC- patients’ groups (Appendix A).

### 3.5. Receptor Expression Profile of NKT-Like and T Cells

NKp46, KIR2DL1/S1, and CD161 positive NKT-like cells were comparable between peripheral blood and PF of HGSOC patients. However, NKp46 and CD161 positive NKT-like cell subsets were reduced in tissue specimens. KIR2DL1/S1+NKT-like cells were increased in tumor specimens when compared with their level in PF (Figure 4a–c). KIR2DL2/L3+NKT-like cells were reduced in PF than their level in peripheral blood (Figure 4d). NKp46, DNAM-1, and CD161 positive T cells were low in tumor specimens than PF. However, these T cell subsets were comparable in peripheral blood and PF (Figure 4e–g). No significant differences were found in the receptor expression data of PF resident NKT-like and T cells between ASC+ and ASC- groups (Appendix A).

### 3.6. Reduced Soluble MICB and PVR in Peritoneal Fluid of HGSOC Patients

We have measured the level of soluble ligands such as MICA, MICB, PVR, and B7-H6 in control serum, patient serum, and PF from these patients. The soluble level of MICA in the PF was comparable with the serum of the patient and HC (Figure 5a), while soluble MICB and PVR levels were low (Figure 5b,c), and the B7-H6 level was high (Figure 5d). The ligand data of ASC+ and ASC- patients groups indicated that the ASC+ group of patients had higher levels of the NKG2D ligand MICB in their PF (Figure 5f), while no difference was observed in other ligands (Figure 5e,g,h).

### 3.7. Elevated Levels of Cytokines in Peritoneal Fluid of HGSOC Patients

Procartaplex immunoassay analysis revealed that IL-2, IL-5, IL-6, IL-8, IL-10, IL-15, IL-27, IFN-γ, TNF-α, and GM-CSF were significantly elevated in PF of HGSOC patients in comparison to their levels in serum samples as well as with that of HC. Furthermore, serum levels of cytokines such as IL-5, IL-6, IL-10, and TNF-α were increased significantly, while levels of IL-2 and IL-8 were reduced significantly in these patients compared to their levels observed in the serum of HC (Figure 6a–j). Disrupted cytokine level in the patient’s serum and PF was further confirmed in the correlation matrix. We have found that cytokine level in the PF was highly correlated in comparison with the patients and control serum (Figure 6k–m). However, no significant difference was observed in the cytokine levels in the PF of ASC+ and ASC- patients groups (Appendix A).

### 3.8. Differential Immune Profile of Chemo Response Score 1 (CRS1) and Chemo Response Score 2/3 (CRS2/3) Group of Patients

Disease-free survival analysis based on chemo response score was carried out. We have found that median disease-free survival in the CRS1 group was 18 months versus 28 months in the CRS2/3 group (HR 1.51; 95% CI: 0.66–3.43, *p* = 0.295), indicating better disease-free survival in CRS2 group, though not significantly different (Figure 7a). Immune monitoring of both groups revealed that the total NKT-like cells were increased in the peripheral blood of the CRS2/3 group (Figure 7b). However, the expression of receptors on NKT-like cells was comparable (Data not shown). Circulatory KIR2DL2/L3/S3+CD56+^Dim^ NK cells and CD161+CD56+^Dim^NK cells (Figure 7c–d), as well as KIR2DL2/L3/S3+CD56+Dim NK cells and DNAM-1+CD56+DimNK cells in the PF, were also increased in the CRS2/3 group of patients (Figure 7e,f). In contrast, KIR2DL2/L3/S3+T cells were decreased in tumor specimens of the CRS2/3 group of patients (Figure 7g). Furthermore, the soluble level of IL-6 in PF and MICA in the serum of the CRS2/3 group of patients was reduced (Figure 7h,i).

## 4. Discussion

Increased infiltration of TILs, composition, and phenotype in various cancers determines the antitumor response and clinical outcome. The expression profile mapping of co-inhibitory receptors LAG-3, TIM-3, and PD-1 in paired PF and tissue specimens highlights the mechanism to reverse the T cell dysfunction which may be used to develop immunotherapeutic approaches [15]. The positive prognosis for patients with solid tumors may be predicted by high levels of NK cell markers in tumor tissues [16]. These reports indicated the importance of our detailed immunophenotyping of CD56^Bright^NK, CD56^Dim^NK, NKT-like, and T cells in peripheral blood, PF, and tissue specimens from the same set of ovarian cancer patients at the time of interval debulking surgery. Investigating the role of soluble factors in PF and tumors also highlights the crucial association of these factors with CA-125 [15]. Thus, the measurement of soluble ligands and cytokine profiles in paired serum and PF samples may further highlight their association with ovarian cancer. A recent study highlights the presence of a relatively mature phenotype of NK cells (CD56^Dim^ KIR+CD57+CD16^High^) in PF of high-grade peritoneal carcinomatosis [17].We have found a reduced frequency of immature (CD56^Bright^)NK cells in the PF of HGSOC patients. CD3+T cells were gradually decreased from peripheral blood to PF to tumor tissue. CD56^Dim^NK, NKT-like, and T cells were reduced more prominently in tumor tissue, suggesting that the tumor site is largely immune-deprived and may be associated with poor prognosis as revealed in the large ovarian tumor cohort study that T cell exclusion in tumors is associated with poor prognosis [18].There was a significant infiltration of CD56^Dim^NK and CD3+T cells in PF of patients with disseminated tumor cells (ASC+). However, receptor expression on NK, NKT-like, and T cells have no significant difference in the ASC+ and ASC- patients group indicating dissemination of tumor cells did not have a marked influence on receptor expression on these immune cells in PF. In addition, disseminated tumor cells may influence the accumulation of immune cells in PF. CD56^Bright^NK cells are the major producers of cytokines such as IFN-γ, TNF-α, GMCSF, IL-10, and IL-13 depending on the stimulation [19]. It was also reported that CD56^Bright^NK cells have the capacity for antiviral and antitumor functions [20]. As a result, a reduction of CD56^Bright^NK cells positive for NKG2C, NKG2D, DNAM-1, and KIR3DL1 in PF may affect their functional characteristics. Surprisingly, NKp46+CD56^Dim^NK cells were increased in PF. The development of a multifunctional NK cell engager targeting NKp46 on tumor-infiltrated NK cells enhances protective tumor immunity [21]. Thus, high expression of NKp46 specifically on tumor-associated mature NK cells may provide a unique opportunity for such type of immunotherapeutic intervention in ovarian cancer. A gradual decrease of DNAM-1 may indicate the chronic exposure of tumor-associated and tumor-infiltrated NK cells which may represent an important mechanism of immune escape and tumor progression and may also compromise synapse formation and IFN-γ production [22,23]. Furthermore, reduced expression of NKG2C, NKG2D, CD161, KIR2DL2/L3/S3, and KIR2DL1/S1 on CD56^Dim^NK cells likely reflects the exhaustion of NK cells in tumor specimens. Imbalanced receptor expression may lead to NK cell dysfunction in TME. The chemotherapy response score is used to determine the prognosis of the disease and classify ovarian cancer patients based on the CRS score into a high risk of relapse [24]. However, we did not find a significant difference in progression-free survival based on the CRS score. CD8+ T cell infiltration was shown to play an important role in chemo-radiotherapy response in cisplatin-treated HNSCC patients [25]. Interestingly, in our study, immune parameters such as KIR2DL2/L3/S3 +CD56^Dim^NK cells in the blood, PF and tissue correlated with the CRS score. DNAM-1+ and CD161+CD56^Dim^ NK cells were also correlated with CRS scores in PF and blood, respectively. It has been reported that immune subset score and ratio correlate with complete remission, persistent disease and complete response [26]. Thus, immune parameters play an important role in predicting response to chemotherapy. Therefore, KIR2DL2/L3/S3 along with CD161 and DNAM-1 receptor expression could be an important immunological determinant for chemotherapy response in HGSOC patients. Moreover, NKT-like cells expressing prominent functional receptors NKp30, NKp46, and CD161 were reduced, whereas KIR2DL2/L3/S3+NKT-like cells were increased in the PF. It is also reported that NKT-like cells participate directly or indirectly in anti-tumor immunity [27]. Similar to NK cells, NKT-like cell activation depends on the balance between the stimulatory and inhibitory signals generated by the engagement of NCRs and KIRs to their cognate ligands [27]. Thus, dysregulated receptor expression on tumor-associated NKT-like cells may impair their anti-tumor response. Further investigation with CD3+T cells highlighted the reduced level of NKp46, DNAM-1, and CD161 positive T cells in tumor tissue. These NK cell receptors have a crucial function on T cells and act as a co-stimulatory molecule on CD8+ T cells for efficient stimulation by non-professional antigen-presenting cells and optimal activation of effector functions of CD8+T cells in tumor tissue, whereas CD161+CD8+ T cells show long-term memory, cytotoxic potential, drug efflux capacity, and are the candidate of choice for adoptive cell transfer therapy [28,29]. Therefore, comparative downregulation of DNAM-1, NKp46, and CD161 on tissue-infiltrated T cells may compromise their functional properties to induce the immune system and anti-tumor potential in ovarian cancer.

Shedding of cognate ligands for activating NKG2D receptors in PF correlates with disease severity and NK cell dysfunction in endometriosis [30]. However, we report that MIC-B and PVR were significantly reduced in the PF of HGSOC, although grouping patients based on the dissemination of tumor cells into ASC+ and ASC- groups revealed themselves to be highly soluble MIC-B in ASC+ patients, which indicates that the dissemination of tumor cells may directly contribute to the shedding of MIC-B in PF. Though soluble MICA was neither significantly altered in the serum and PF of patients nor the ASC+ and ASC- patients group, its level was correlated with the CRS score in the serum of these patients. Metalloprotease-mediated shedding of other soluble ligands such as B7-H6 was reported previously [31]. In our study, soluble ligand B7-H6 was increased in PF which may act as a decoy molecule to downregulate the expression of NKp30 and enhance the immune escape of tumor cells [32]. However, despite highly soluble B7-H6, NKp30 expression was not reduced relatively in PF. A detailed investigation may highlight the more complex role of soluble B7-H6 in the peritoneal cavity.

Investigation of the cytokine profile in paired serum and PF revealed that all the cytokines were high in PF, which indicates a cytokine storm-like condition in the peritoneal cavity. An elevated level of cytokines in the peritoneal cavity was also reported by another group [15]. A previous report indicates that tumor-associated lymphocytes in the peritoneal cavity are more competent cytokine producers than their counterparts in peripheral blood [33]. Thus, increased cytokines in PF could be the result of the efficient cytokine-producer property of lymphocytes in the peritoneal cavity. However, we did not find any prognostic association of these cytokines in the present study. Interestingly, among all the elevated cytokines in PF, only IL-6 was correlated with the CRS score. The role of IL-6 in stromal inflammation and response to chemotherapy is also reported previously [34]. There is accumulating evidence of IL-6 secreted by the cancer-associated fibroblasts that plays an important role in chemoresistance [35]. Thus, the reduction of IL-6 in PF of the CRS2/3 group of patients could be another important soluble determinant for chemo response in HGSOC. However, in our cohort, there was no significant difference in the level of cytokines between the ASC+ and ASC- group of patients, indicating that tumor cells may not be affecting the cytokine profile but do affect the soluble ligands in PF.

## 5. Conclusions

In ovarian cancer, tumor tissues are more immune-deprived than the peritoneal cavity and blood. Dissemination of tumor cells in PF affects survival and contributes to soluble ligands of NK cells. Increased NKp46+CD56^Dim^NK cells may be targeted by multifunctional engagers in the peritoneal cavity. Immune parameters such as KIR2DL2/L3/S3 along with CD161 and DNAM-1 receptor expression could be an important immunological determinant for chemotherapy response in HGSOC patients. Soluble ligands MIC-Band PVR were reduced; whereas B7-H6 was increased in PF. Soluble immune parameters such as MICA were correlated with the CRS score in the serum of these patients. Interestingly, among all the elevated cytokines in PF, only IL-6 was correlated with the CRS score. In conclusion, the study highlights the differential immune profile in blood, PF, and tumor specimens. The correlation of soluble parameters and NK cell receptors with chemo response score may add knowledge to understand the differential anti-tumor immune response in peripheral blood and TME to develop effective treatment modality.

## Figures and Tables

**Figure 1 vaccines-10-02121-f001:**
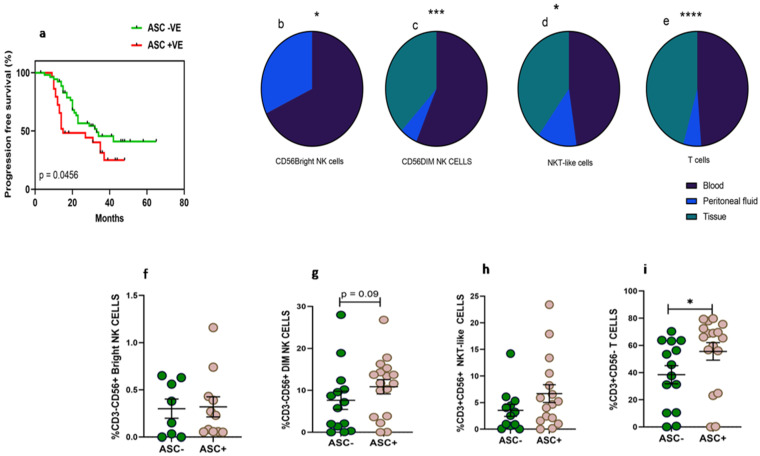
(**a**) Association of fluid cytology status with progression-free survival in HGSOC patients using Kaplan–Meier analysis; Wilcoxon matched-pairs signed rank test for frequency of immune cells (**b**) CD56^Bright^NK cells; (**c**–**e**) mixed effect-ANOVA analysis in peripheral blood, peritoneal fluid, tissue specimens, respectively (**c**) CD56^Dim^NK cells, (**d**) CD3+CD56+NKT-like cells and (**e**) CD3+T cells. A Mann–Whitney U-test (**f**–**i**) for frequency of immune cells in ascites positive (ASC+) and ascites negative (ASC-) for disseminated tumor cells * *p* < 0.05; *** *p* < 0.001; **** *p* < 0.0001.

**Figure 2 vaccines-10-02121-f002:**
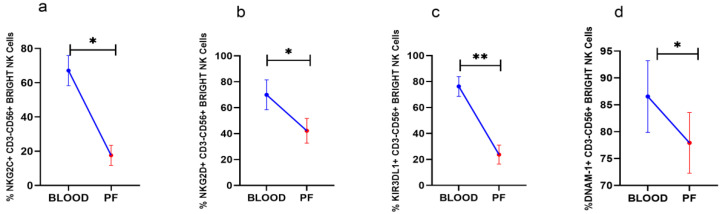
Wilcoxon matched-pairs signed ranked test (**a**–**d**) for receptor expression profile of CD56^Bright^NK cells in peripheral blood and peritoneal fluid (PF) * *p* < 0.05; ** *p* < 0.01.

**Figure 3 vaccines-10-02121-f003:**
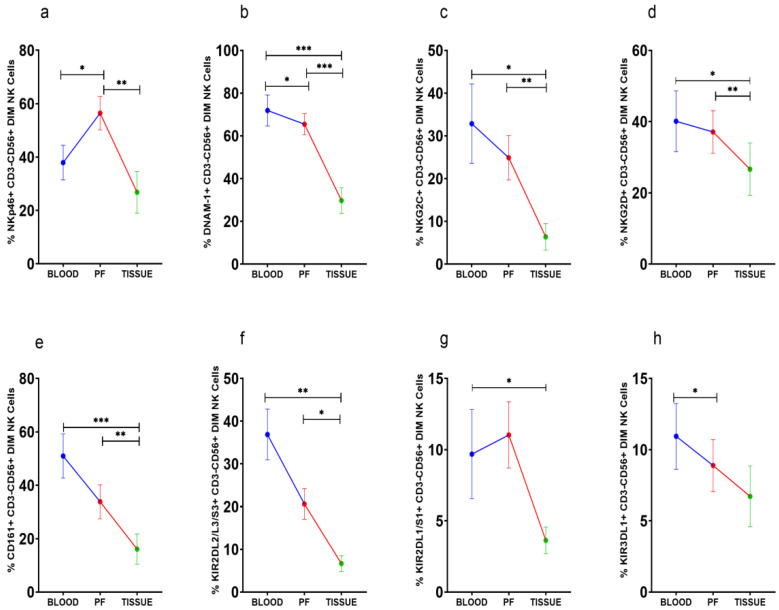
Wilcoxon matched-pairs signed ranked test (**a**–**h**) for receptor expression profile of CD56^Dim^NK cells in peripheral blood, peritoneal fluid (PF), and tissue specimens * *p* < 0.05; ** *p* < 0.01; *** *p* < 0.001.

**Figure 4 vaccines-10-02121-f004:**
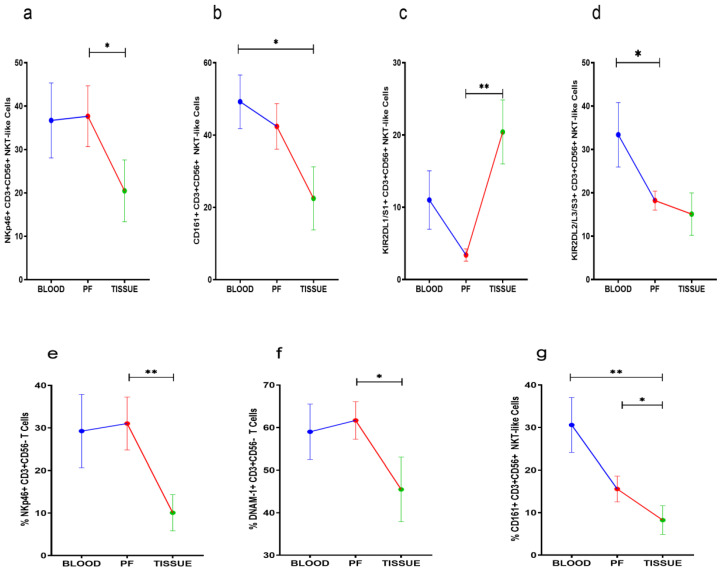
Wilcoxon matched-pairs signed ranked test (**a**–**d**) for receptor expression of NKT-like cells and (**e**–**g**) for receptor expression profile of T cells in peripheral blood, peritoneal fluid (PF), and tissue specimens * *p* < 0.05; ** *p* < 0.01.

**Figure 5 vaccines-10-02121-f005:**
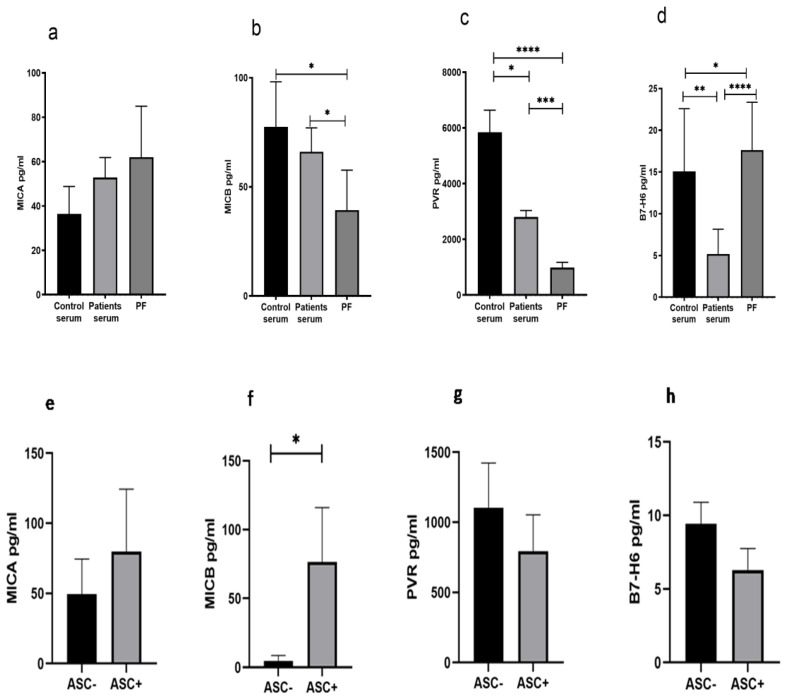
Soluble ligand level (**a**–**d**) in Control serum, Patient’s serum, and Peritoneal Fluid (PF). Soluble ligands level (**e**–**h**) in ASC+ and ASC- peritoneal fluid (PF) sample * *p* < 0.05; ** *p* < 0.01; *** *p* < 0.001; **** *p* < 0.0001.

**Figure 6 vaccines-10-02121-f006:**
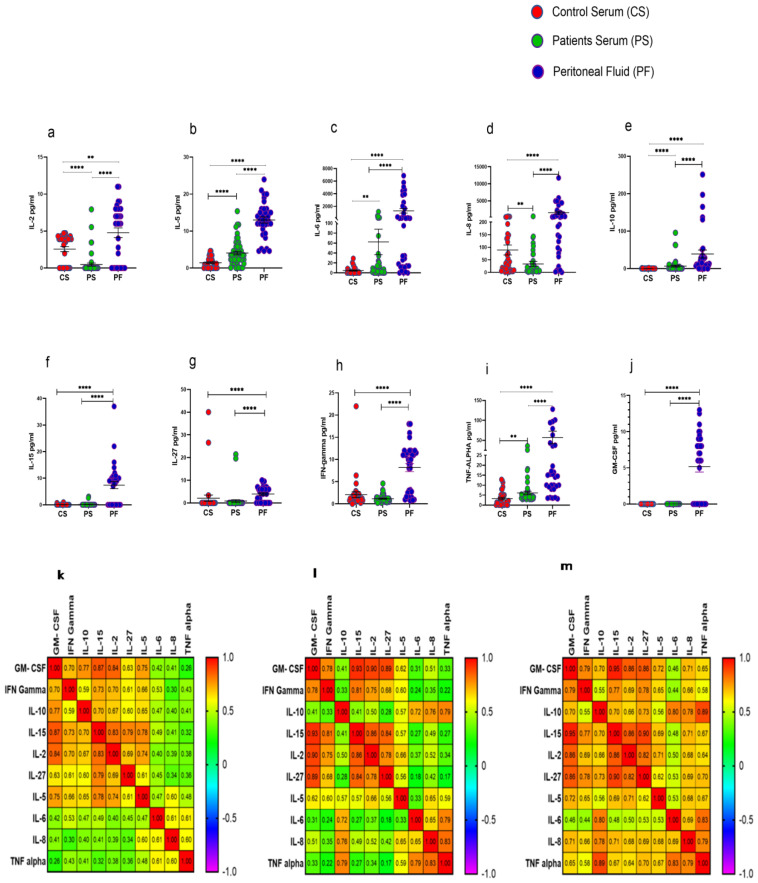
Cytokine profile in Control serum (CS), Patients serum (PS), and peritoneal fluid (PF) (**a**–**j**); Spearman Correlation matrix of cytokine profile in (**k**) healthy serum, (**l**) patient’s serum, and (**m**) peritoneal fluid. ** *p* < 0.01; **** *p* < 0.0001.

**Figure 7 vaccines-10-02121-f007:**
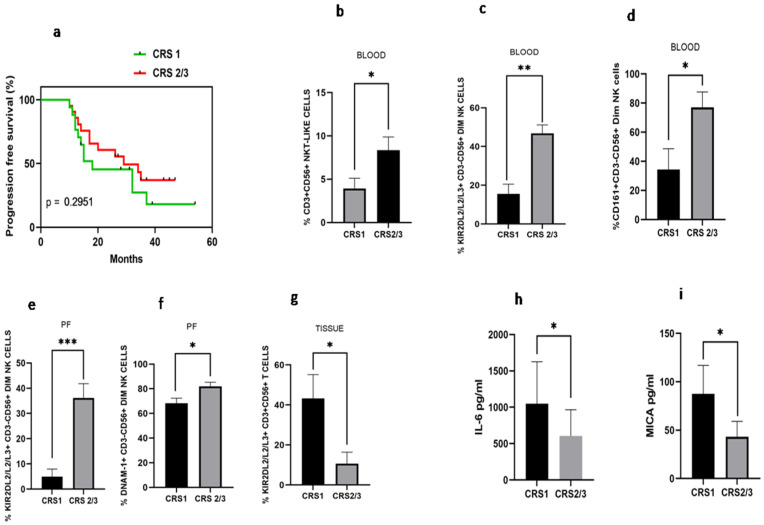
(**a**) Progression-free survival with Chemo response score 1 (CRS 1) and Chemo response score 2/3 (CRS 2/3); (**b**) NKT-like cells; (**c**) KIR2DL2/L3/S3+CD3-CD56+^Dim^ NK; (**d**) CD161+CD3+CD56+^Dim^ NK cells in peripheral blood; (**e**) KIR2DL2/L3/S3+CD3-CD56+^Dim^ NK and (**f**) DNAM-1+ CD3-CD56+^Dim^ NK cells in Peritoneal fluid; (**g**) KIR2DL2/L3/S3+CD3+CD56-T cells in tissue; (**h**) IL-6 level in peritoneal fluid; and (**i**) serum MICA level of patients with CRS1 and CRS 2/3 group.* *p* < 0.05; ** *p* < 0.01; *** *p* < 0.001.

**Table 1 vaccines-10-02121-t001:** Detailed characteristics of the enrolled patients.

Patients Enrolled	Patients Characteristics	Healthy Controls
	*n* = 33	*n* = 34
Median age (range) in years	56.5 (32–76)	51 (35–67)
Chemotherapy at the time of enrollment	3 cycles (Paclitaxel+Carboplatin)	Not applicable
Stage		Not applicable
II	4 (12.1%)	
III	21 (63.4%)	
IV	8 (24.2%)	
Grade		Not applicable
High	33 (100%)	
Low	0	
Unknown	0	
Co-morbidity		
Yes	18 (54.5%)	Arbitrary Healthy No active chronic infection Diabetes, Hypertension, etc.
No	10 (30.3%)	
Unknown	5 (15.1%)	
Fluid cytology		Not applicable
Positive	18 (54.5%)	
Negative	15 (45.4%)	
Unknown	0	
Lymph node metastasis		Not applicable
Positive	16 (48.4%)	
Negative	10 (30.3%)	
Unknown	7 (21.1%)	
Chemo response score (CRS)		Not applicable
CRS 1	14 (42.4%)	
CRS 2	16 (48.4%)	
CRS 3	3 (0.09%)	
CA-125 Median (Range)		Not applicable
Pre-treatment	665.5 (125.4–27,428) units/mL	Not applicable
Post-treatment	37.94 (6.9–679.5) units/mL	

## Data Availability

The datasets generated during and/or analyzed during the current study are presented in the manuscript as figures and tables and will also be available from the corresponding author on reasonable request.

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
