# Peer review of "Immune Profile of Blood, Tissue and Peritoneal Fluid: A Comparative Study in High Grade Serous Epithelial Ovarian Cancer Patients at Interval Debulking Surgery"

_vaccines, 2022, doi:10.3390/vaccines10122121_

Round 1

Reviewer 1 Report (Previous Reviewer 2)

The authors performed immune profiling of blood, tissue, and ascites at the time of interval debulking surgery in cases of high-grade serous epithelial ovarian cancer.

The effort of careful sampling is commendable.

But what can be said after comparing these three types of specimens in the same disease? It would be impossible to mention the characteristics of the disease in this subject disease without at least a similar comparison with other types of ovarian cancer or with benign diseases of the ovary.

Or, if sampling over time is significant, what would happen if we grouped the samples by outcome and compared them?

Although interesting as an experiment, the results of this study are simply "we tried this" and we do not know what we want to seek from it as a medical science.

I understand the efforts of sampling, etc. very well, so I hope that this research will be expanded and developed to make discoveries that will lead to the characterization of this disease or a cure for it.

Author Response

Dear Reviewer,

We have revised the manuscript with minor and major corrections in the abstract, introduction, material and methods, results as well as in the discussion, wherever required as per your suggestions. Major corrections are in red fonts in the word file.

Thank you so much for the important suggestions to revise the manuscript.

Please see the attachment for the details.

With regards

Dr Jayanti Mania-Pramanik

Reviewer 2 Report (Previous Reviewer 1)

Your changes and comments are acceptable.

Author Response

Dear Reviewer,

Thank you so much for the important suggestions to revise the previous version of manuscript and accepting the changes.

With regards

Dr Jayanti Mania-Pramanik

Round 2

Reviewer 1 Report (Previous Reviewer 2)

The modifications according to the reviewers' comments seemed to be fine, now.

This manuscript is a resubmission of an earlier submission. The following is a list of the peer review reports and author responses from that submission.

Round 1

Reviewer 1 Report

Overall, the manuscript, in particular introduction and discussion, is much to long and hard to read. Moreover, presentation of results could be more clearly too. 

The idea behind your study is good, but there are some shortcomings. First of all it is necessary to differentiate patients receiving neoadjuvant therapy in this with regress, stable disease, and progressive disease. I am surprised to read that you also have patients in FIGO stage I receiving neoadjuvant therapy, because potential resectable disease should be completely debulked. 

For preparation of single cell suspensions for flow analysis it is necessary to have a control section of the tissue before mincing. Otherwise you don't have a control about the percentage of tumor vs stroma. Has this been done?

My major point of concern, however, is that you have 45% of patients with negative ascites. HGSCs themselves frequently behave like an inflammatory cell, producing a lot of pro-inflammatory cytokines. Therefore, to get a real picture it is necessary to look on these two groups (ASC+ versus ASC-) separately. 

Author Response

Dear Reviewer,

We are thankful to the editor and reviewers for their critical and useful suggestions. We have revised the whole manuscript considering all the comments and suggestions made by the reviewers. The point-wise responses to the reviewers’ comments are attached here.

Please let us know if further clarifications are required to improve the submitted article.

We thank you for your comments/suggestions to improve and allow us to revise the manuscript.

Reviewer 2 Report

We compared immune profiles in peripheral blood, ascites, and tumor tissue in cases of high-grade serous epithelial ovarian cancer undergoing interval debulking surgery.

The experiments carried out seem to have been carried out carefully.

However, in the first place, what do you want by measuring immunocompetent cells and cytokines in blood, ascitic fluid, and tissue, which have different natures? It is natural to be different, and nothing will come out of comparing all cases.

For some factor, for example, it would be interesting to discover that the prognosis, responsiveness to treatment, and pathological condition differ depending on the level in blood and ascites fluid, but is such an approach being taken? ?

The value of collecting various data is acknowledged. Therefore, I would like you to reconsider what and how you want to clarify using them.

Author Response

Dear Reviewer,

We are thankful to the editor and reviewers for their critical and useful suggestions. We have revised the whole manuscript considering all the comments and suggestions made by the reviewers. The point-wise responses to the reviewers’ comments are attached herewith for your consideration.

Please let us know if further clarifications or modifications are required.

We thank you for your comments/suggestions to improve and allow us to revise the manuscript.

With regards

Dr Jayanti Mania-Pramanik
